# Effect of the Combination of Torsional and Tensile Stress on Corrosion Behaviors of Biodegradable WE43 Alloy in Simulated Body Fluid

**DOI:** 10.3390/jfb14020071

**Published:** 2023-01-28

**Authors:** Bowen Wang, Wei Gao, Chao Pan, Debao Liu, Xiaohao Sun

**Affiliations:** 1School of Materials Science and Engineering, Tianjin University of Technology, Tianjin 300384, China; 2Beijing Chunlizhengda Medical Instruments Co., Ltd., Beijing 300384, China; 3National Demonstration Center for Experimental Function Materials Education, Tianjin University of Technology, Tianjin 300384, China; 4Tianjin Key Laboratory for Photoelectric Materials and Devices, Tianjin 300384, China

**Keywords:** degradation behavior, magnesium alloy, stress-corrosion cracking, tensile loading, torsional loading

## Abstract

The real physiological environment of the human body is complicated, with different degrees and forms of loads applied to biomedical implants caused by the daily life of the patients, which will definitely influence the degradation behaviors of Mg-based biodegradable implants. In the present study, the degradation behaviors of modified WE43 alloys under the combination of torsional and tensile stress were systematically investigated. Slow strain rate tensile tests revealed that the simulated body fluid (SBF) solution could deteriorate the ultimate tensile stress of WE43 alloy from 210.1 MPa to 169.2 MPa. In the meantime, the fracture surface of the specimens tested in the SBF showed an intergranular corrosion morphology in the marginal region, while the central area appeared not to have been affected by the corrosive media. The bio-degradation performances under the combination of torsional and tensile stressed conditions were much more severe than those under unstressed conditions or single tensile stressed situations. The combination of 40 MPa tensile and 40 MPa torsional stress resulted in a degradation rate over 20 mm/y, which was much higher than those under 80 MPa single tensile stress (4.5 mm/y) or 80 MPa single torsional stress (13.1 mm/y). The dynamic formation and destruction mechanism of the protective corrosion products film on the modified WE43 alloy could attribute to the exacerbated degradation performance and the unique corrosion morphology. The dynamic environment and multi-directional loading could severely accelerate the degradation process of modified WE43 alloy. Therefore, the SCC susceptibility derived from a single directional test may be not suitable for practical purposes. Complex external stress was necessary to simulate the in vivo environment for the development of biodegradable Mg-based implants for clinical applications.

## 1. Introduction

In recent years, biodegradable Mg-based alloys have attracted increasing research interest and attention due to their lightweight, high specific strength, good biocompatibility, and biodegradability, and have exhibited great prospects for biomedical applications [1,2]. More importantly, Mg is highly abundant in the human body and is essential for the metabolism in many biological mechanisms as a co-factor for many enzymes [3,4]. Further, the release of Mg^2+^ ions derived from the degradation process of Mg-based implantation contributes to the healing process and the regeneration of tissue [5]. Further, the biodegradable Mg-based bone scaffold was designed and developed for bone regeneration [6]. Among numerous Mg-based alloys, WE43 alloy containing Rare-earth elements resulted in improved corrosion resistance and enhanced mechanical performance [7,8]. Moreover, modified WE43 showed good biocompatibility and osteoconductivity with no signs of cytotoxicity, contributing to the successful market introduction of bone screws for the treatment of hallux valgus and fractures of small bones in the European Union [9,10,11].

However, Mg-based implants suffer from stress corrosion cracking (SCC) under service in the human body [12,13]. In recent years, the research regarding the SCC behavior of different Mg-based alloys, including AE44, AZ, and Mg-Re alloys, were extensively reported [14,15,16,17]. The physiological environments, including the corrosive media in the human body and complex loads applied to biomedical implants caused by the daily life of the patients, may exacerbate the corrosion progress and result in a loss of mechanical integrity and in a high hydrogen release rate, which the bone tissue was hard to accommodate [18,19]. Bobby Kannan et al. reported that all the ZE41, QE22, Elektron 21 (EV31A), and AZ80 alloys were susceptible to SCC in a corrosion media at a very low-stress level [20]. Choudhary et al. confirmed that applied stress exacerbates the degradation behavior of the aluminium-free ZX50, WZ21, and WE43 alloys in SBF solution [21]. Therefore, the SCC could cause too fast degradation and eventually result in catastrophic failure of WE43 implant during surface.

An in-depth understanding of SCC, its mechanisms, and its evolution is the prerequisite for depression and resistance to SCC. The SCC susceptibility generally depended on applied stress, corrosion media, and alloys compositions. Miller reported that the SCC susceptibility of Mg-based alloys could increase with the Zn content and could be suppressed by RE element addition [22]. Although modified WE43 alloy were available in clinical practice over the last decade and much literature investigated the SCC and corrosion fatigue behavior of WE43 alloy, the fundamental mechanism governing SCC is blurry. Atrens et al. reported that the threshold of SCC was 0.8 times the yield strength for WE43B alloy [23]. Zheng et al. investigated the corrosion fatigue behavior of WE43, and found a much lower fatigue strength tested in the SBF solution than those in the air [2]. However, the applied stress in these studies was limited to a single direction, but the biodegradable implant could endure multi-direction stress in daily life. The influence of the combination of the action of corrosive media and complex external stress on the degradation behavior of WE43 alloy is unclear and should be elucidated.

In the present study, the impact of the combination of torsional and tensile stress on the stress corrosion cracking (SCC) susceptibility of a modified WE43 alloy in a simulated body fluid (SBF) solution was investigated via a slow strain rate tensile (SSRT) test. The dynamic degradation rate and the residual strength after immersion under various stress were also evaluated to reveal the impact of applied stress on the degradation behavior and mechanical integrity of biodegradable WE43 alloy.

## 2. Materials and Methods

### 2.1. Preparation of Modified WE43 Alloys

The raw materials used in this study are pure commercial Mg (purity, 99.995%), pure Zn particles (purity, 99.995%), Mg-30Y, Mg-30Nd, Mg-30Zn, and Mg-25Ca intermediate alloys. A self-modified furnace with high-speed agitation and ultrasonication accessories was used to manufacture the modified WE43 alloys under a protective atmosphere of SF_6_ and N_2_ mixture. The pure magnesium ingots were melted at 993 K, and then added the intermediate alloy at 1023 K, followed by stirring at a speed of 4000 r/min for 20 min. Afterward, the WE43 alloy was cast at 1023 K. Thereafter, solid solution treatment was performed at 798 K for 8 h, then quenched at 333 K. Subsequently, hot extrusion was executed at 633 K with an extrusion rate of 1 mm/s, and an extrusion ratio of 42.25:1. Finally, WE43 alloy bars with a diameter of 10 mm were obtained.

### 2.2. Phase Analysis and Microscopic Observation

The specimen was polished with 320#, 800#, 1500#, and 3000# sandpaper in turn followed by ultrasonication in ethanol. X-ray diffractometer (XRD, SmartLab 9kW, Rigaku, Tokyo, Japan) was used to investigate the phase constitution of cleaned specimens with a scan range of 10° to 90° and a scanning speed of 8°/min. After general metallurgical polishing, the experimental specimens were etched in a picric acid solution composed of Bitter acid 2.75 g, anhydrous ethanol 45 mL, ice acetic acid 2.5 m, and deionized water 5 mL. Optical microscope (OM, Olympus GX51, Olympus, Tokyo, Japan) and field emission scanning electron microscope (FE-SEM, Quanta FEG 250, Thermo Fisher Scientific, Waltham, MA, USA) were used for microscopic observation, and X-ray energy dispersive spectrometer (EDS, Thermo Fisher Scientific, Waltham, MA, USA) was employed to analyze the elemental composition.

### 2.3. In Vitro Static and Dynamic Immersion Test

The in vitro static and dynamic immersion tests were performed in SBF solution at 310 K. The specimen for the static immersion tests was executed in the biochemical incubator (CN-40A, As one, Tokyo, Japan), while the constant temperature water bath oscillator at 310 K with 60 oscillations per minute was used for the dynamic immersion test. The fluctuation of the immersion temperature was less than 0.5 K. The immersion periods were set as 1 day, 3 days, 7 days, 14 days, 21 days, and 30 days. Three parallel specimens are tested at each time period. The SBF solution was refreshed every two days. Inductively coupled plasma atomic emission spectrometry (ICP-OES, Vista-MPX, Thermo Scientific, Waltham, MA, USA) was used to determine the concentration of magnesium ions in the immersed solution. The composition of corrosion products was determined by EDS. The corrosion product layer on the surface was removed by using the proportionally configured chromium acid washing solution (200 g/L CrO_3_, 20 g/L Ba(NO_3_)_2_, 10 g/L AgNO_3_). The weight loss of immersed specimens was measured by a digital balance (FA224, Lichen, Shanghai, China) with a 0.1 mg accuracy. The corrosion rate of the material was calculated by the following formula according to ASTM-G31-72 [24].
(1)CR=(k×w)÷(A×t×D)

In the formula, *CR* is corrosion rate (mm/y), coefficient *k* is a constant and equal to 8.76 × 10^4^, *w* is weight loss (g) after the immersion test, *A* is specific surface area (cm^2^), *t* is immersion period (h), and *D* is the apparent density of the experimental specimens (g·cm^−3^).

### 2.4. Slow Strain Rate Tensile (SSRT) Test in Air and SBF Solution 

According to ASTM-G129-Y2021 [25], the test was carried out in the laboratory inert atmospheric environment and the simulated body fluid at a constant temperature of 310 K. The schematic diagram of the SSRT test in the study was shown in Figure 1. Before the SSRT test in corrosive media, the surface of the specimen outside the gauge area was blocked by the Teflon insulating tape. The self-modified biomechanical testing machine (DDL020-50, Sinotest equipment, Beijing, China) with the corrosion tank equipped with a constant temperature heating device was used. The strain rate was 1 × 10^−6^ s^−1^, and the simulated body fluid was replaced every 24 h. Dog-bone shape specimens with a gauge length of 25 mm and gauge diameters of 5 mm were used in the SSRT test. At least three parallel specimens are set for each specimen.

### 2.5. Immersion Test in SBF Solution under Torsional and Tensile Stress

The degradation behavior under torsional and tensile stress was tested by the self-modified biomechanical testing machine. In order to simulate the applied stress due to the daily activity of the patient, the selected load values were set under the yield strength of intact WE43 specimens. In this study, experiment specimens were immersed in SBF solution and 310 K under single tensile stress, single torsional stress, or the combination of tensile and torsional stress. The single tensile stress of 20 MPa, 40 MPa, and 80 MPa, the single torsional load of 40 MPa and 80 MPa, and the combined stresses of 20 MPa (tensile) + 40 MPa (torsional) and 40 MPa (tensile) + 40 MPa (torsional) was applied on the experimental specimens. The corrosion media was SBF solution and the immersion period was 3 days. The fluctuation of immersion temperature and applied force was less than 0.5 K and 1 N, respectively. The simulated body fluid is replaced every 24 h. After the immersion, the corrosion morphology and corroded substrate (after the removal of corrosion products) were observed by FE-SEM. Afterward, the residual strength of the specimen was evaluated by the tensile test with the strain rate of 1 × 10^−3^ mm/s. At least, three duplicates were tested and the stress-free condition was employed as the reference.

## 3. Results and Discussions

### 3.1. Microstructure and Phase Constitution of Modified WE43 Alloy

Figure 2a showed OM images of as-cast WE43 alloy. A typical dendritic microstructure was observed. As exhibited in Figure 2b, significant grain refinement occurred during the hot extrusion process, and equiaxed grains were confirmed in as-extruded WE43 alloy. The mean grain size decreased from 371 ± 22 μm in as-cast WE43 alloy to 13 ± 1.2 μm in as-extruded counterparts. According to Figure 2c, many second-phase particles with a submicron size were observed, and the second-phase particles were discontinuously distributed along the hot extrusion direction, implying the movement of the second-phase particles during the hot extrusion process. As shown in Figure 2d,e, EDS analysis revealed the enrichment of Nd and deficiency of Mg in these second-phase particles, indicating the formation and precipitation of Mg-Nd second phases. Further, the X-ray profile of the as-extruded WE43 alloy was displayed in Figure 2f, and specific peaks from α-Mg matrix, Mg_12_Nd second phase, and Mg_41_Nd_5_ were determined, illustrating the second phases observed in the SEM image was Mg_12_Nd and Mg_41_Nd_5_. These fine second-phase particles could provide a pinning effect inhibiting grain growth and dislocation movement, thereby refining the microstructure and enhancing the mechanical strength [26]. It is worth noting that the absence of the Mg-Y seconding phase may be due to the high solid solubility of Y in the Mg matrix (12.3 wt%) [27]. 

### 3.2. Degradation Behavior during the Static and Dynamic Immersion Tests

Figure 3a–d exhibited corrosion morphology after static immersion tests for 1 day, 7 days, 14 days, and 30 days. When the immersion period was 1 day, plate-like morphology with many crevices was observed and the corroded surface of WE43 alloy. With the extension of immersion time, the cracks between the plate-like were gradually reduced. In the meantime, the corrosion product particles formed, and gradually covered the whole corrosion surface. After 30 days of immersion (Figure 3e), the crevices almost disappeared and a thick corrosion product layer was deposited on the surface of the WE43 alloy. According to the EDS analysis (Figure 3f), the corrosion products were mainly composed of Mg, Ca, O and P, indicating the formation of MgO and calcium−phosphorus compound. MgO could gradually transform to Mg(OH)_2_ in an aqueous environment and due to the presence of Cl^−^ in the SBF solution, the insoluble Mg(OH)_2_ could transform to soluble MgCl_2_ for long immersion periods. In other words, the formation of the insoluble calcium phosphorus compound may restrict the corrosion progression, while the MgO was unreliable as the protective layer in chlorine-containing SBF solution. More importantly, the calcium−phosphorus compound possessed excellent biocompatibility and osteogenic capability, which suggested increased bioactivity and biocompatibility during the degradation process of WE43 alloy [28,29,30].

Figure 4a–d exhibited corrosion morphology after dynamic immersion tests for 1 day, 7 days, 14 days, and 30 days. As compared with the results of static immersion tests, the dynamic immersion tests also resulted in many crevices after 1 day of immersion. However, some precipitations were confirmed on the dynamic corroded surfaces, while a relatively smooth surface was formed in the counterparts immersed in the static environment. With the increased immersion time, the cracks were gradually disappeared. After 14 days of dynamic immersion, a relatively integrated corrosion products layer with limited shallow cracks was confirmed, suggesting the reduced corrosion rate. Nevertheless, the corrosion layer seems to become fragmentary after 30 days of dynamic immersion. It is supposed that the integrity of the protective layer was deteriorated by the oscillation, and a dynamic equilibrium of formation and destruction of the protective layer was achieved. Figure 4e,f proved that similar chemical composition was measured in the corrosion products formed during the static and dynamic immersion test, although the morphology was different. Thus, the dynamic environment may show a limited impact on the kind of corrosion product but have a great influence on the formation of the protective layer.

Figure 5 exhibited the corroded substrates of WE43 specimens for the static and dynamic immersion test after the removal of corrosion products. In the initial stage of the static immersion test and dynamic immersion test (1 day), the surface corrosion degree of the two kinds of magnesium alloy specimens is low, and the grinding marks generated in the specimen preparation process are still visible. In the early stage of static immersion WE43 magnesium alloy (7 days) appeared with some reticular structure, which may be due to the micro galvanic corrosion induced by the second phase containing rare earth elements, which is more resistant to corrosion than the Mg matrix. A previous study investigated the Volta potential of the second phase containing rare earth elements and Mg matrix in WE43 alloy via scanning Kelvin probe force microscope. A relatively high potential was confirmed in the secondary phase, while the Mg matrix showed lower surface Volta potential [31]. From thermodynamic interpretation, a micro-galvanic couple consisting of the second phase containing rare earth elements as potential cathode and the Mg matrix as potential anode could be formed, thereby accelerating the dissolution of the Mg matrix during the degradation process. After 30 days of static immersion, some areas showed intergranular corrosion morphology, and others areas exhibited a relatively homogeneous corrosion morphology. On the other hand, a total intergranular corrosion morphology with deep corrosion pits was confirmed in the dynamic immersed WE43 alloy. It is supposed that localized corrosion could provide more corrosion products and facilitate the deposition of the calcium−phosphorus compound, and these corrosion products could obstruct the corrosive media to a fresh surface thereby slowing down the degradation progression. However, the dynamic environment inhibited the corrosion product accumulation, and expose the Mg matrix to the SBF solution, and the previously generated small pitting pits will continue to expand to the deep part of the specimen, thus resulting in a typical localized corrosion morphology.

Figure 6a illustrates the degradation rates determined by the static and dynamic immersion tests with different immersion periods. The degradation rates determined by static immersion tests were continuously reduced with the increasing immersion periods, due to the gradual formation of the protective corrosion layers with relatively high integrity. On the contrary, the degradation rates derived from the dynamic immersion tests decreased first and then maintained at a relatively stable value of about 2.8 mm/y. Further, in the initial stage of corrosion (1 day and 3 days), dynamic immersion contributed to a higher corrosion rate, but static immersion resulted in faster corrosion in the immersion range of 7–14 days. When the immersion period was over 14 days, the corrosion rates determined by the dynamic immersion test were higher than those derived from static immersion tests once again. Figure 6b showed the Mg ion concentration in the SBF solution during the immersion tests. During the first 10 days, the SBF solution of immersion static and dynamic immersion tests contained the same Mg ion content. However, when the immersion period was over 10 days, the Mg ion contents increased with the extension of immersion times in dynamic immersion, while the Mg ion concentration in static immersion tests was maintained at a relatively low level in the immersion period of 10 to 30 days. These results are also attributed to the gradual formation of the protective corrosion products layer and the destructive effect of the dynamic environment on the protective layer [32,33]. Above all, the dynamic environment will cause a higher degradation rate and a faster Mg ion release than its static counterparts. Due to the rapid metabolic rate in the human body, dynamic immersion tests should be more suitable to evaluate the degradation behavior of metallic biomaterials in the real physiological environment.

### 3.3. SSRT Tests in Different Environments

Figure 7 illustrated the Nominal stress–strain curves derived from the SSRT test of WE43 alloy in air and SBF solution. The stress corrosion cracking sensitivity factor *I*_SCC_ is calculated from elongation after fracture (E), section shrinkage (S), ultimate tensile strength (UTS), and fracture total time (FTT). The relevant data are summarized in Table 1. All the UTS, FTT, S, and E determined in the SBF solution were lower than those measured in the air, indicating that the corrosive environment was very harmful to the mechanical performance of biomedical WE43 alloy. The *I*_E_ and *I*_S_ were 0.68 and 0.41, respectively. These results illustrated that the corrosive environment not only deteriorate the mechanical strength but also showed an unfavorable effect on the ductility of WE43 alloy. Further, the WE43 alloy fabricated in this study showed a similar *I*_UTS_ value to the literature value. We also found that the WE43 alloy showed relatively high *I*_UTS_ and *I*_E_ values by comparing the stress corrosion cracking sensitivity factors of various Mg-based alloys [7,21,34], which means that the SCC susceptibility of WE43 was lower than other Mg-based alloys. Therefore, from the viewpoint of SCC resistance and biosafety, WE43 alloy was more suitable than other Mg alloys as load-bearing biodegradable implants. In order to further explore the failure mechanism of WE43 during the SSRT test, the fracture surfaces of two specimens in the air and the SBF solution were observed by scanning electron microscopy.

Figure 8 exhibited the fracture morphology of WE43 alloy obtained by SSRT tests in the air at 310 K. Micro voids, ductile fracture morphology, and cleavage fracture morphology could be observed. The microvoids could be caused by the pinning effect of the hard second phase. The second phase particle could act like an immobile core and some crevice formed around the hard core. On the other hand, the ductile fracture occurred in the central areas of the fracture surface, while the cleavage fracture morphology was confirmed in the margin region. Similar results were reported in some previous research [35,36,37]. Furthermore, the cleavage plane was very obvious in Figure 8 taken from the direction perpendicular to SSRT orientation. That may be due to the Mg-based alloys, as HCP metals possess a limited active slip system, thereby resulting in a partial cleavage fracture morphology. It is worth mentioning that some streamline perpendicular to the tensile direction were observed on the specimens’ surface, which is close to the fracture surface. These results indicated the initial morphology of the cleavage fracture, the fracture could progress along the streamline and finally result in failure.

Figure 9a is a macrograph of the fracture surface after SSRT measurement in SBF solution at 310 K. The fracture surface is generally divided into three areas: the ductile fracture zone in the center area, the cleavage fracture zone at the edge of the ductile fracture area, and the immersion zone at the outermost layer of the specimen. Similar to the specimens measured in the air, ductile fracture morphology and cleavage fracture morphology co-existed on the fracture surface as shown in Figure 9b,c. However, according to Figure 9d, an immersion zone in the marginal region of the fracture surface with evident corrosion pits was observed, suggesting the corrosion progression in these areas. Moreover, Figure 9e showed a cleavage fracture surface with many corrosion patterns, which illustrated that cleavage fracture and corrosion were two concurrent phenomena during the SSRT in the SBF solution. As exhibited in Figure 9f, some intergranular corrosion morphology was observed in the region relatively far from the gauge surface, implying the corrosive media can enter the Mg matrix through cracks, crevices, streamlines, and corrosion pits [38,39,40]. Therefore, the corrosive SBF solution could facilitate the intergranular corrosion on the margin region and these corrosion areas induce reduced ductility, triggering the transition from ductile fracture to cleavage fracture. As a result, both the mechanical strength and ductility were decreased in the corrosive media, as compared with the reference materials in the air.

### 3.4. In Vitro Immersion Test under the Combination of Tensile and Torsional Stress

Figure 10 showed the corroded surface of the gauge area of WE43 specimens after immersion in SBF solution at 310 K under single tensile stress, single torsional stress, the combination of tensile, and torsional stress. Similar morphology was confirmed in all experimental specimens, but some deep corrosion pits were observed in the specimens applied by the combination of tensile and torsional stress. Therefore, the combination of tensile and torsional stress could facilitate the pitting corrosion, thereby deteriorating the mechanical performance during the service of WE43 implants.

Figure 11a–c displayed degradation rates determined by the weight loss method after immersion in SBF solution at 310 K under single tensile stress, single torsional stress, and the combination of tensile and torsional stress. The degradation rates significantly increased the increasing applied stress. Moreover, the single torsional stress exhibited more impact than a single tensile test. More importantly, the combination of 40 MPa tensile and 40 MPa torsional stress resulted in a degradation rate over 20 mm/y, which was much higher than those under 80 MPa single tensile stress (4.5 mm/y) or 80 MPa single torsional stress (13.1 mm/y). In other words, complex stress with a low-stress level may result in a more significant effect on the degradation rate of WE43 implants than unidirectional stress with high-stress levels. Figure 11d–f exhibited residual strength determined by the tensile test after immersion in SBF solution under single tensile stress, single torsional stress, and the combination of tensile and torsional stress. As compared to the stress-free reference, the residual strength of the specimen decreased with the increasing applied stress, despite the different kinds of stress. Interestingly, the residual mechanical strength with 80 MPa single tensile stress, 80 MPa single torsional stress, and the combination of 40 MPa tensile and 40 MPa torsional stress was 202.5 MPa, 211.3 MPa, and 201.3 MPa, respectively. These results indicated that although single tensile stress or single torsional stress could accelerate the degradation process of biodegradable WE43 alloy in SBF solution, the combination of tensile stress and torsional stress greatly increased the corrosion rate, indicating that the comprehensive influence of the corrosive environment and complex applied stress could be significant to the degradation behavior of WE43 alloy and should be further researched in the near future to ensure the biosafety and mechanical integrity of Mg-based biomedical implant.

Two possible mechanisms were responsible for the accelerated corrosion progress via applying external stress. One is that when the external loading was applied to WE43 alloy, the surface free energy was elevated, and the solid binding energy decreased with the increasing applied stress [41]. The reduced binding energy caused the reduction of reaction activation energy, thereby facilitating the dissolution of the Mg matrix. Further, the transformation from MgO to Mg(OH)_2_ and from Mg(OH)_2_ to MgCl_2_ in SBF solution was also promoted by the elevated reactive activity. Several previous articles reported that stress could facilitate the nucleation and growth of pitting pits [42] after the formation of the small corrosion pits caused by high reaction activity. The stress could concentrate at these corrosion pits and then further accelerate the corrosion progression, resulting in the expansion and coalescence of corrosion pits. The larger pit diameter and the higher applied stress exacerbate the stress concentration at the bottom of the corrosion pits. Therefore, the Gibbs free energy increase, and the activated dissolution of the Mg alloy surface in pits would be accelerated. As a result, the deeper and larger corrosion pits were confirmed at the complex stressed condition (Figure 10). Another mechanism was that the external loading could induce and promote microcrack propagation on the surface of the protective corrosion product layer [43]. Additionally, the hydrogen produced by the corrosion of Mg could destruct the bonding at the crack tip, and then cause hydrogen embrittlement and crack growth, as well as accelerate further corrosion [44]. Multi-directional stress may cause different corrosion progression orientations, implying that the integrity of the corrosion product layer could be further destructed and collapsed under complex external stress. Above all, the accelerated Mg dissolution and reduced integrity of the corrosion product layer were the two main reasons responsible for the biocorrosion behavior of WE41 alloy under applied stress. 

In this study, the combination of torsional stress and tensile stress was applied to WE43 alloy to disclose the corrosion behavior under complex external stress. However, the stress level and stress orientation could be random and continuously fluctuate in the actual situation, which means that the corrosion fatigue process under multi-directional stress was closer to the actual situation than those of this study. The corrosion fatigue behavior of WE43 alloy will be investigated under multi-directional stress in the near future. Further, the in vivo study may directly verify the biosafety of WE43 alloy under complex stress and could be another research orientation. Moreover, the computational simulation regarding the strain distribution under complex enteral stress [45] and the corrosion progress induced by different stress levels [46] could be another direction for future research due to its faster results and lower cost as compared with the experimental investigation.

## 4. Conclusions

This paper mainly studied the influence of the dynamic environment and complex applied stress on the bio-corrosion behavior of modified WE43 alloy. The results demonstrated that a relatively dense protective layer was formed on the surface of WE43 alloy after static immersion tests, indicating restricted degradation progress. However, the dynamic immersion environment reduced the integrity of the protective corrosion product layer which resulted in increased Mg ion concentration (about 160 μg/mL) in SBF solution and exacerbated the corrosion process. SSRT tests revealed that the SBF solution could deteriorate the mechanical integrity of WE43 alloy. The corrosion environment caused the UTS to decrease from 210.1 MPa to 169.2 MPa and the elongation reduced from 41.5% to 28.3%. Meanwhile, an intergranular corrosion morphology in the marginal region was confirmed on the fracture surface of the specimens tested in SBF solution, but the central area appeared not to have been affected by the corrosive media. Although single tensile stress or single torsional stress could accelerate the degradation process of biodegradable WE43 alloy in SBF solution, the combination of tensile and torsional stress induces significant localized corrosion and greatly increased corrosion rate (over 20 mm/y), indicating the complex stress caused by daily activity could considerably impact the degradation behavior of WE43 biomedical implant. Further research including the corrosion fatigue behavior under multi-directional load, the in vivo degradation behavior under complex external stress, and the development of the anti-SCC Mg-based alloy will be performed in the future.

## Figures and Tables

**Figure 1 jfb-14-00071-f001:**
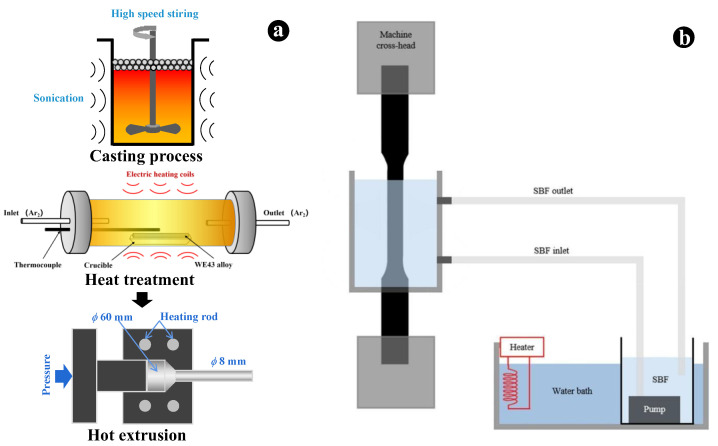
Schematic diagrams of (**a**) materials preparation and (**b**) slow strain rate tensile test in simulated body fluid solution.

**Figure 2 jfb-14-00071-f002:**
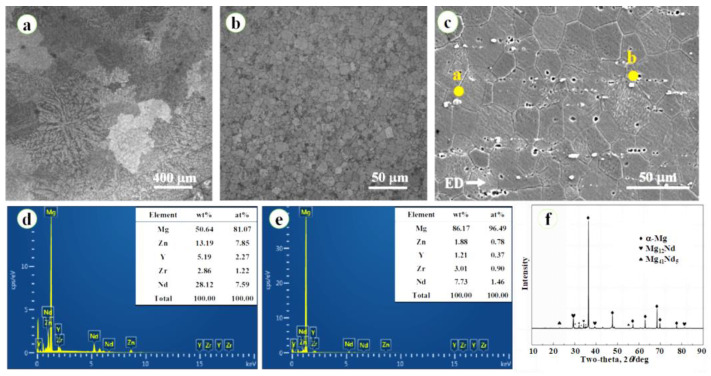
Optical microscope (OM) images of (**a**) as-cast and (**b**) as-extruded WE43 alloy; (**c**) scanning electron microscope (SEM) image and (**d**,**e**) corresponding energy dispersive spectrometer (EDS) point analysis results; (**f**) X-ray profile of as-extruded WE43 alloy.

**Figure 3 jfb-14-00071-f003:**
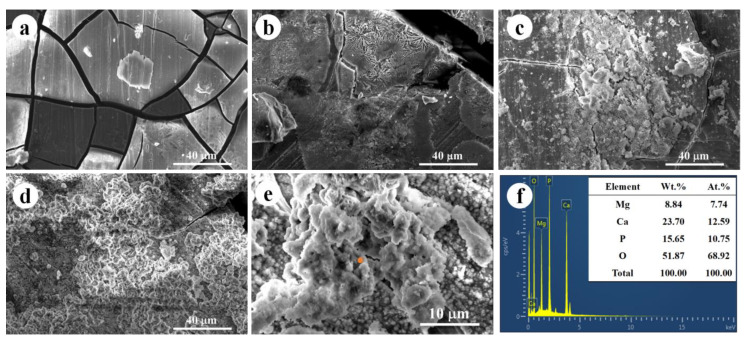
(**a**–**d**) Corrosion morphology after static immersion tests for 1 day, 7 days, 14 days, and 30 days; (**e**) high magnification image and (**f**) corresponding EDS evaluation of corrosion morphology after 30 days immersion.

**Figure 4 jfb-14-00071-f004:**
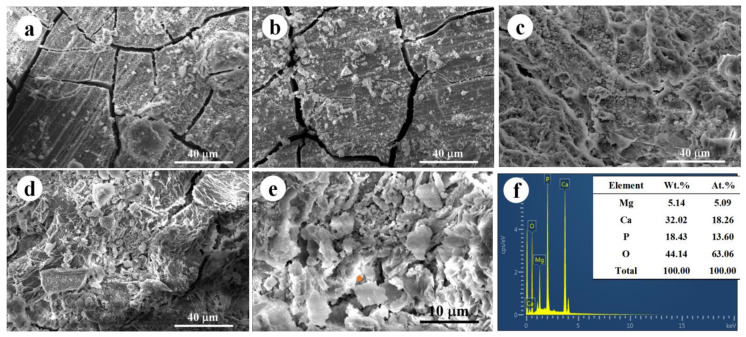
(**a**–**d**) Corrosion morphology after dynamic immersion tests for 1 day, 7 days, 14 days, and 30 days; (**e**) high magnification image and (**f**) corresponding EDS evaluation of corrosion morphology after 30 days immersion.

**Figure 5 jfb-14-00071-f005:**
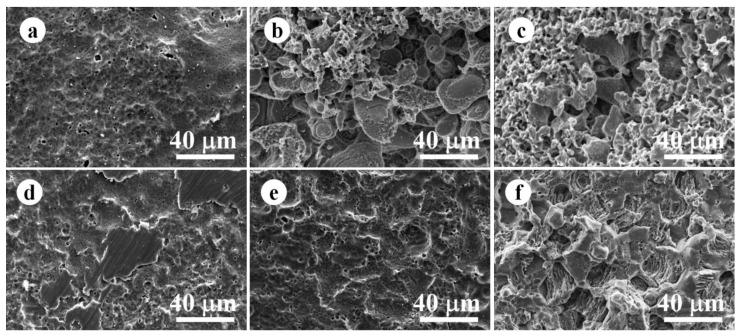
(**a**–**c**) Morphology of WE43 alloy after removal of corrosion products on the 1, 7, and 30 days of static state; (**d**–**f**) Morphology of WE43 alloy after removal of corrosion products on the 1, 7, and 30 days of dynamic state.

**Figure 6 jfb-14-00071-f006:**
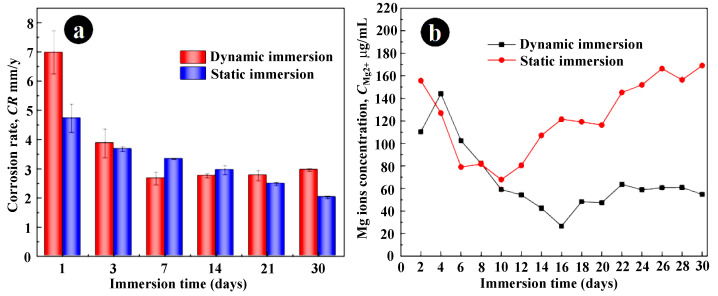
(**a**) Degradation rate evaluated via static and dynamic immersion test; (**b**) the Mg^2+^ concentration in SBF solution during the immersion test at 310 K.

**Figure 7 jfb-14-00071-f007:**
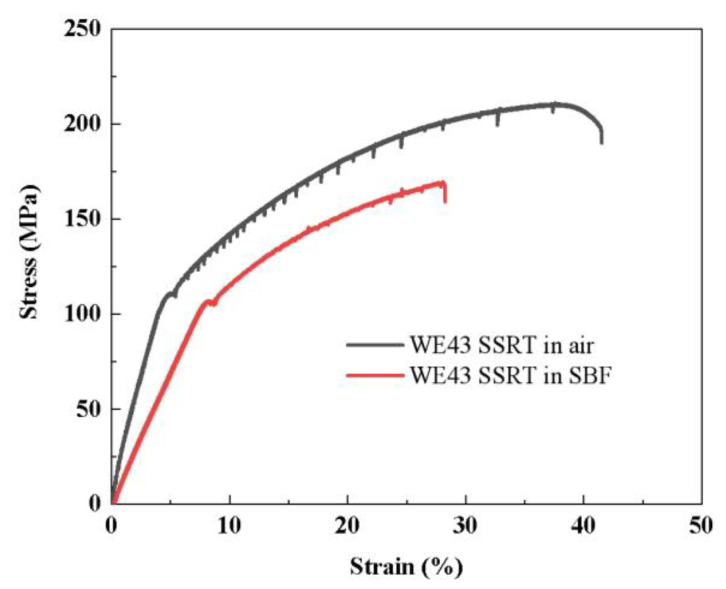
Nominal stress–strain curves derived from slow strain rate tensile (SSRT) test of WE43 alloy in air and SBF solution.

**Figure 8 jfb-14-00071-f008:**
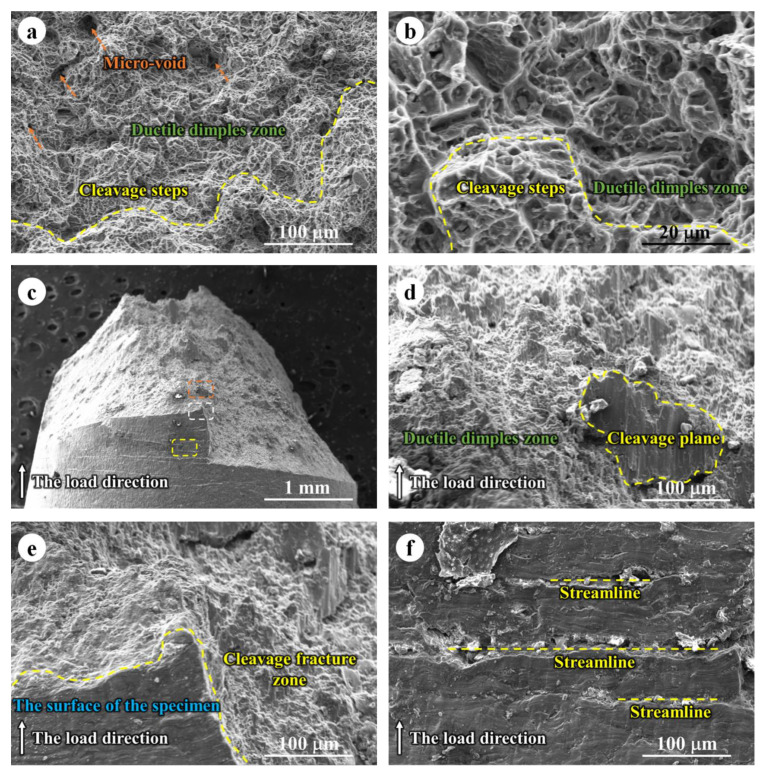
Fracture observation of WE43 alloy after SSRT test in the air at 310 K: (**a**,**b**) SEM images observed from the orientation parallel to the tensile direction; (**c**–**f**) SEM images observed from the orientation parallel to the tensile direction; Magnified images taken from (**d**) orange dashed frame, (**e**) white dashed frame, and (**f**) yellow dashed frame in (**c**).

**Figure 9 jfb-14-00071-f009:**
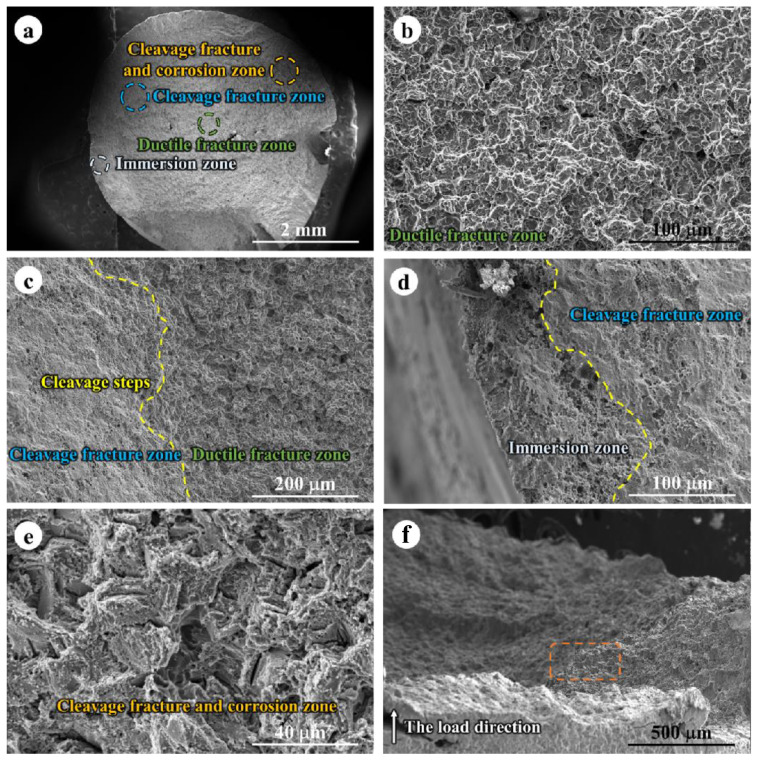
Fracture observation of WE43 alloy after SSRT test in the air at 310 K: (**a**) low magnificent image of fracture morphology; magnified images from (**b**) ductile fracture zone (green dotted circle), (**c**) intermediate area between cleavage fracture zone and ductile fracture zone, (**d**) intermediate area between cleavage fracture zone and immersion zone and (**e**) immersion zone (white dotted circle); (**f**) SEM images observed from the orientation perpendicular to the tensile direction; Orange dashed frame indicates the typical cleavage fracture morphology.

**Figure 10 jfb-14-00071-f010:**
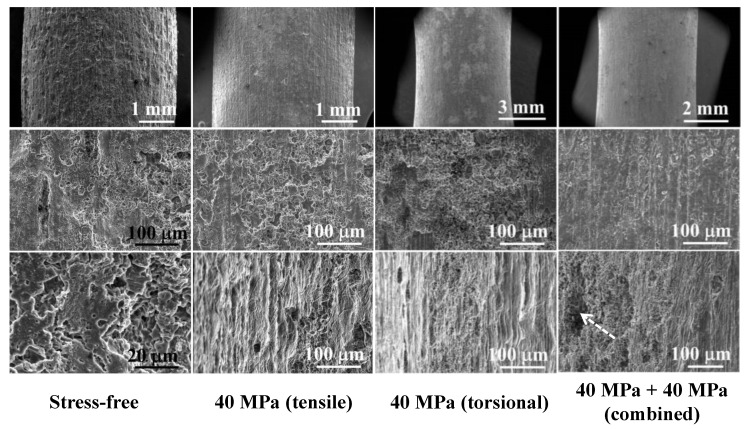
The corroded surface of the gauge area of WE43 specimens after immersion in SBF solution at 310 K under single tensile stress, single torsional stress, and the combination of tensile and torsional stress (white dashed arrow indicates the significant corrosion pit).

**Figure 11 jfb-14-00071-f011:**
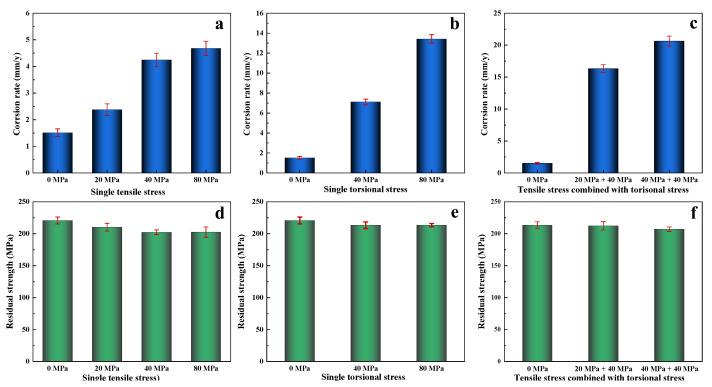
Degradation rates were determined by the weight loss method after immersion in SBF solution at 310 K under (**a**) single tensile stress, (**b**) single torsional stress, and (**c**) the combination of tensile and torsional stress; residual strengths were determined by the tensile test after immersion in SBF solution under (**d**) single tensile stress, (**e**) single torsional stress, and (**f**) the combination of tensile and torsional stress.

**Table 1 jfb-14-00071-t001:** Summary of SSRT test results and stress corrosion cracking sensitivity factors of various Mg-based alloys.

Materials	Environment	UTS (MPa)	FTT (h)	E (%)	S (%)	*I* _UTS_	*I* _FTT_	*I* _E_	*I* _S_	Ref.
WE43	Air	210.1	146.3	41.5	40.5	0.80	0.54	0.68	0.41	This work
	SBF	169.2	78.5	28.3	16.5					
ZX50	Air	352.5		21.2		0.73		0.18		[21]
	SBF	257.0		3.8						
WZ21	Air	242.5		28.1		0.68		0.27		[21]
	SBF	166.2		7.7						
WE43	Air	263.0		16.9		0.80		0.47		[21]
	SBF	211.0		8.0						
ZK21	Air					0.19		0.62		[32]
	SBF									
AZ31	Air	256.3		24.5		0.91		0.25		[7]
	SBF	233.3		6.1						

## Data Availability

The raw and processed data required to reproduce these findings cannot be shared at this time as the data also forms part of an ongoing study.

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
