# Peer review of "Effect of the Combination of Torsional and Tensile Stress on Corrosion Behaviors of Biodegradable WE43 Alloy in Simulated Body Fluid"

_jfb, 2023, doi:10.3390/jfb14020071_

Round 1

Reviewer 1 Report

The manuscript is about the effect of the combination of torsional and tensile stress on corrosion behaviors of biodegradable WE43 alloy in simulated body fluid. It is possible to say that the writing language is very good and the presentation is done well. It is determined that results are technically correct.

The application of tensile and SBF tests together is a very demanding application and the authors have succeeded in it.

Please consider following comments:

1) At the end of the introduction, it is strongly recommended authors to identify the research gap clearly in the literature.

In research gap, it will be seen that study has an important novelty.

2) Adequate literature research has not been done. The introduction needs to be expanded.

3) Most of the images have high resolution. It will be better to increase resolution of Fig 2f. also visibility of letters of Fig2f.

Figures 11a, b and c are blurred. Please increase quality of the graphics.

4) In abstract and conclusion parts, it is recommended to give also numerical results.

Author Response

We would like to thank the reviewers and the editors for the careful and thorough reading of this manuscript and for the thoughtful comments and constructive suggestions, which helped to improve the quality of this manuscript. Please find the point-by-point responses in the attachment file.

Reviewer 2 Report

In the present research authors investigated the corrosion Behavior of Biodegradable WE43 Alloy in Simulated Body Fluid after different loading regimes. High reactivity of magnesium for its use in human body is still a major concern towards wider usage in biomedicine since the evolution of Hydrogen as a result of cathodic reaction should be kept under the control.  Presented research deals with a real life application of WE43 magnesium alloy.  Though the research exhibits many results, there are still some questions in the present state. 

Questions: 

1. Lines (95-96). For better explanation, I suggest authors could write down the formula by which was calculated corrosion rate.

2. Line (156). Authors stated that Mg, Ca, O and P were present on the immersed surfaces indicating formation of MgO. What happens with MgO after being exposed in aqueous environment. Is there any transformation after MgO being hydrated for longer exposure time? 

3. Lines (193-194). Authors are suggested to explain why is secondary phase containing rare earth elements more resistant to corrosion than Mg matrix. The suggestion is oriented on thermodynamic of electrochemical corrosion. 

4. Lines (221-222; 234-235). Missunderstanding in Figure 6b.  The lines ( 221-222) describes the Mg ion concentration in the SBF solution during the immersion tests not the pH evolution as is stated in (234-235). 

5. Lines (350-351). The following sentence is not complete: (1) Relatively dense protective layer was formed on the surface of WE43 alloy after...

6. Lines (352-353). How Mg2+ ion release influences pH values ?   

Author Response

(The authors gave the same response as above.)

Reviewer 3 Report

1.      The abstract should be broadened to give additional quantitative results.

2.      Please conclude your abstract with a "take-home" message.

3.      Reorder keywords based on alphabetical order.

4.      Nothing truly unique in its current state. Because of the lack of novel, the current study looks to be a replication or modified study. The authors must describe their novel in detail. This work should be rejected owing to a major issue.

5.      Previous study related needs to explain in the introduction section consisting of their work, their novelty, and their limitations to show the research gaps that intend to be filled in the present study.

6.      Line 30-41 the authors explaining about magnesium biodegradable materials for prospect of biomedical application. Recent relevant study published by MDPI explaining the content from Putra et al. that need to be incorporated as follows: Level of Activity Changes Increases the Fatigue Life of the Porous Magnesium Scaffold, as Observed in Dynamic Immersion Tests, over Time. Sustainability 2023, 15, 823. https://doi.org/10.3390/su15010823

7.      In order to improve the reader's understanding of the materials and methods section simpler, the authors could provide a figure that clarify the workflow of the current study rather than only the predominant text as it currently appears.

8.      It also is needed to include more information on tools, such as the manufacturer, the country, and the specification.

9.      The inaccuracy and tolerance of the experimental equipment used in this inquiry are critical details that must be included in the article.

10.   A comparative assessment with similar previous research is required.

11.   Overall, discussion in the present article is extremely poor. The Authors must extend their discussion and make a comprehensive explanation. Just not simply mention the results with brief explanation.

12.   Please include the limitation of the present study, it is missing.

13.   Elaborate the conclusion as a form of paragraph, not point by point as present form.

14.   In the conclusion, please explain the further research.

15.   The reference needs to be enriched from the literature published five years back. MDPI reference is strongly recommended.

16.   The manuscript needs to be proofread by the authors since it has grammatical and language issues.

17.   It is mandatory to provide a graphical abstract after revision is submission system.

Author Response

(The authors gave the same response as above.)

Round 2

Reviewer 3 Report

I have further comments to response the authors revised version:

1.      In line 434, the authors use “we”, please make it into passive to more scientific explanation. Also, apply it to all of the manuscript.

2.      Since the present study performs experimental investigation, potential further study with computational simulation needs to be explained that offer some advantages compared to experimental for investigate metal implant, such as faster results and lower cost. Suggested reference is needed as follows: The Effect of Bottom Profile Dimples on the Femoral Head on Wear in Metal-on-Metal Total Hip Arthroplasty. J. Funct. Biomater. 2021, 12, 38. https://doi.org/10.3390/jfb12020038

3.      Figure 11 and 12 is would be better to combined since it is similar form.

Author Response

We would like to thank the reviewers and the editors for the careful and thorough reading of this manuscript and for the thoughtful comments and constructive suggestions, which helped to improve the quality of this manuscript. Point-by-point responses can be found in the attachment file.

Round 3

Reviewer 3 Report

I am recommending the present manuscript to be accepted.